# E-CLEAP: An ensemble learning model for efficient and accurate identification of antimicrobial peptides

Si-Cheng Wang [ORCID]*

School of Statistics and Applied Mathematics, Anhui University of Finance and Economics, Bengbu, China

* 20202523@aufe.edu.cn

## Abstract

With the increasing problem of antimicrobial drug resistance, the search for new antimicrobial agents has become a crucial task in the field of medicine. Antimicrobial peptides, as a class of naturally occurring antimicrobial agents, possess broad-spectrum antimicrobial activity and lower risk of resistance development. However, traditional screening methods for antimicrobial peptides are inefficient, necessitating the development of an efficient screening model. In this study, we aimed to develop an ensemble learning model for the identification of antimicrobial peptides, named E-CLEAP, based on the Multilayer Perceptron Classifier (MLP Classifier). By considering multiple features, including amino acid composition (AAC) and pseudo amino acid composition (PseAAC) of antimicrobial peptides, we aimed to improve the accuracy and generalization ability of the identification process. To validate the superiority of our model, we employed five-fold cross-validation and compared it with other commonly used methods for antimicrobial peptide identification. In the experimental results on an independent test set, E-CLEAP achieved accuracies of 97.33% and 84% for the AAC and PseAAC features, respectively. The results demonstrated that our model outperformed other methods in all evaluation metrics. The findings of this study highlight the potential of the E-CLEAP model in enhancing the efficiency and accuracy of antimicrobial peptide screening, which holds significant implications for drug development, disease treatment, and biotechnology advancement. Future research can further optimize the model by incorporating additional features and information, as well as validating its reliability on larger datasets and in real-world environments. The source code and all datasets are publicly available at https://github.com/Wangsicheng52/E-CLEAP.

## 1 Introduction

Antimicrobial peptides (AMPs) are a class of small molecular peptides that exhibit broad-spectrum antimicrobial activity and have significant potential in combating microbial infections and the development of novel antibiotics [1,2]. Traditional antibiotics face increasing challenges of drug resistance, while AMPs, as a new antimicrobial strategy, offer advantages such as broad antimicrobial spectra, rapid action, low resistance development, and minimal

**Data Availability Statement:** The link is: https://github.com/Wangsicheng52/E-CLEAP.

**Funding:** this study was supported by the China Postdoctoral Science Foundation (2021M691345), and the funders had no role in study design, data

collection and analysis, decision to publish, or preparation of the manuscript.

**Competing interests:** The authors have declared that no competing interests exist.

selective pressure [3]. AMPs not only possess bactericidal properties but also exhibit various biological activities, including immune modulation, anti-inflammatory effects, and wound healing promotion [4]. In comparison to traditional antibiotics, AMPs possess unique characteristics and advantages, making them promising candidates for future antibiotic alternatives [5].

In recent years, with the growing concern about antibiotic resistance, efficient peptide screening has emerged as a crucial research direction in the search for new antimicrobial and anticancer agents [6]. Currently, the screening of AMPs heavily relies on laborious and costly laboratory methods, which are often inefficient [7]. However, with the advancements in computer science and biotechnology, the development of efficient models for screening antimicrobial peptides has become highly meaningful.

Currently, numerous studies have delved into the effective screening of diverse peptides using machine learning and deep learning methodologies. For example, in 2023, Davide et al. crafted a deep learning-based model to forecast the activity and drug properties of therapeutic peptides. This model, leveraging insights from known drug peptides, accurately predicts the activity and attributes of unfamiliar peptide sequences [8,9]. Another noteworthy investigation by Sun et al. in 2017 harnessed deep learning algorithms to formulate precise high-throughput approaches for discerning protein-protein interactions, a pivotal aspect for comprehending protein functions, disease mechanisms, and treatment design [10]. In 2021, Lin et al. devised the AI4AMP model, a deep learning-infused protein encoding technique proficient in precisely anticipating the antibacterial activity of a given protein sequence [11]. In 2018, Bhadra et al. developed the AmPEP model, leveraging the random forest algorithm and amino acid property distribution patterns for antimicrobial peptide prediction [12]. Söylemez et al. (2023) introduced the AMP-GMS model, employing a group-based and score-based methodology for antimicrobial peptide prediction [13]. Lastly, Li et al. (2023) utilized bidirectional long short-term memory (Bi-LSTM) and attention mechanisms to construct the AMPlify model, showcasing remarkable performance in antimicrobial peptide prediction [14].

These studies demonstrate the potential of machine learning, deep learning, and computer-aided design methods in the efficient screening and optimization of other peptides. By leveraging information on peptide sequences, structures, and properties, these methods can enhance the activity, selectivity, and stability of peptides. Future research can further develop and integrate these methods to provide more possibilities and opportunities for the discovery and application of other peptides.

The traditional methods for screening antimicrobial peptides are characterized by low efficiency, necessitating the development of an efficient screening model. Therefore, this study proposes a model based on Ensemble Voting for analyzing the features of antimicrobial peptide AAC sequences and PseAAC sequences. To improve the predictive performance of the extracted features, we employ an ensemble classification of four MLPClassifiers and then aggregate the predictions of each MLPClassifier model through voting [15,16]. The model leverages the advantages of multiple MLP classifiers, enabling it to capture complex nonlinear relationships. By addressing data imbalance using the SMOTE algorithm and accurately assessing model performance through stratified 5-fold cross-validation, our model outperforms traditional approaches in terms of predictive performance and robustness. Additionally, objective evaluation of the model's performance is achieved by drawing ROC curves and calculating AUC values, providing evidence for its superiority in analyzing features of antimicrobial peptide sequences. The workflow of this study is illustrated in **Fig 1**.

Overall, the Ensemble Voting model, through innovative ensemble learning methods, comprehensive feature utilization, data balancing techniques, and comprehensive performance

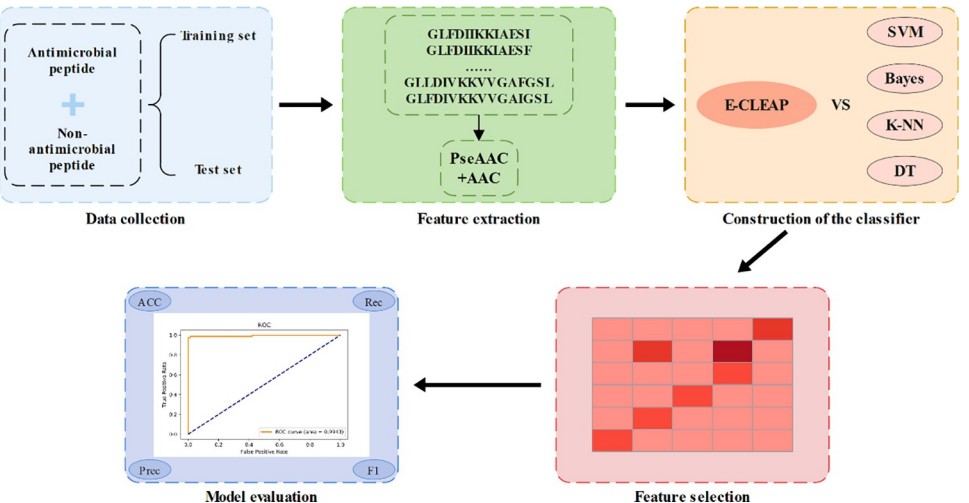

**Fig 1. The flow chart of the data analysis.**

evaluation, has overcome the limitations of traditional antimicrobial peptide screening methods. It proposes a more efficient and superior screening model.

## 2 Materials and methods

### 2.1 Data collection

We collected 1750 experimentally validated antimicrobial peptide samples from the APD3 (https://aps.unmc.edu/about), PlantPepDB (http://14.139.61.8/PlantPepDB/index.php), BaAMPs (https://www.baamps.it/) and BioPepDB (https://bis.zju.edu.cn/biopepdbr/).These samples had unique sequences and were non-synthetic. As a positive control, we extracted 1750 non-antimicrobial peptide samples from the UniProt database (https://www.uniprot.org/). In total, we obtained 3500 peptide sequences as our dataset for this study. To construct the model, we randomly selected 3000 samples as the training set and an additional 500 samples as the test set. It is important to note that we ensured there were no duplicates between positive and negative samples in both the training and test sets, as well as within each sample.

### 2.2 Sequence feature extraction

**2.2.1 AAC characteristic.** The AAC feature is a representation method used to describe the relative abundance of different amino acids in a protein sequence. It calculates the frequency of occurrence of each amino acid in the entire sequence. Given a protein sequence, the AAC feature is calculated using the following formula:

$$AAC(i) = \frac{Frequency\ of\ amino\ acid(i)}{Length\ of\ the\ peptide} \tag{1}$$

$i$ can be any natural amino acid, and AAC has a fixed length of 20 features. We used the SeqIO module from the Biopython library in Python to read protein sequences from the input file. Then, we applied the AAC formula to calculate the AAC feature for each protein sequence.

**2.2.2 PseAAC characteristic.** PseAAC is a commonly used protein sequence feature representation method that captures structural and functional information of sequences. We extract PseAAC features by counting the occurrence frequencies of amino acid fragments of different lengths in the protein sequence.

Specifically, we first calculate the frequency of each amino acid and then generate amino acid fragments of a specified length. We then calculate the occurrence frequency of these fragments. For each pair of amino acids ($aa_1$ and $aa_2$), the calculation formula for PseAAC features is as follows:

$$PseAAC(aa_1, aa_2) = \frac{(x + w)}{[y(aa_1)^* \, y(aa_2) + w^2]}$$

(2)

In the formula, $x$ represents the occurrence count of the dipeptide fragment, $y(aa_1)$ and $y(aa_2)$ represent the frequency of amino acids $aa_1$ and $aa_2$, respectively. During the process, the parameter lambda_value controls the length of the generated amino acid fragments, determining the range of fragment lengths considered. In this case, we set it to 1, considering only amino acid fragments of length 1. The parameter $w$ is a smoothing parameter used to avoid division by zero. By default, we set it to 0.05.

## 2.3 Model construction

**2.3.1 E-CLEAP ensemble model.** The E-CLEAP ensemble model consists of four MLPClassifier models named clf1, clf2, clf3, and clf4. Each MLPClassifier model has different parameters such as hidden layer sizes, random seed, maximum iteration count, learning rate, etc. clf1 and clf4 have two hidden layers with 100 and 50 nodes, respectively. The random seed is set to 420, the maximum iteration count is 5000, the initial learning rate is 0.001, the momentum factor is 0.9, the batch size is 8, the optimization algorithm is Adam, and the activation function is ReLU. clf2 and clf3 have a total of four hidden layers with 100, 100, 50, and 25 nodes, respectively. Other parameters are the same as clf1 and clf4.

The E-CLEAP ensemble model combines the predictions of each MLPClassifier model through voting. Specifically, each model provides probability predictions for the two classes based on the test samples. Using the soft voting mechanism (voting = 'soft'), probabilities are weighted averaged based on the model's weights. The final classification result is determined by the decision from the weighted average probabilities. The model framework is illustrated in **Fig 2**.

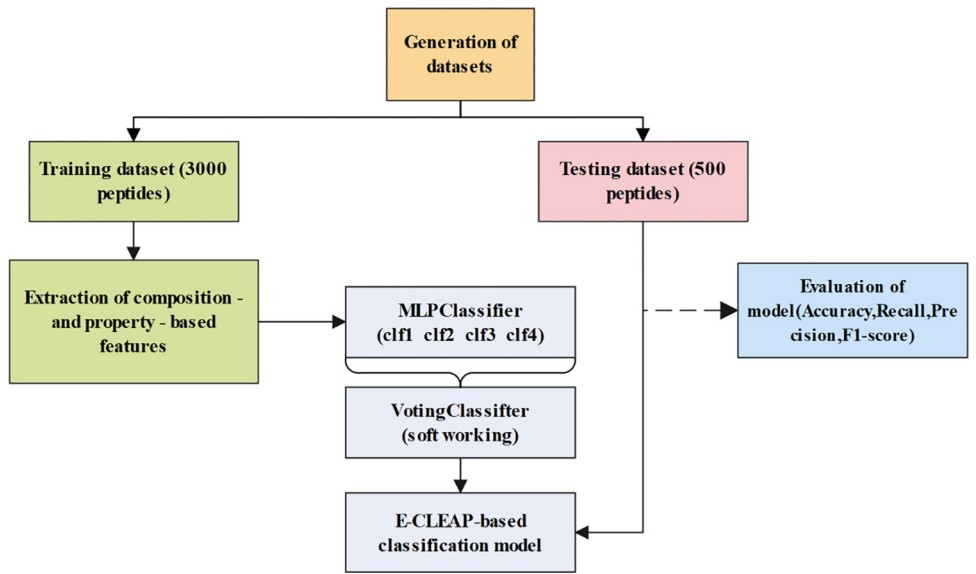

**Fig 2. Ensemble model architecture diagram.**

Voting mechanisms come in two main forms. Hard Voting: Each model classifies the test samples, and the final classification result is the class with the most votes. This is suitable when the models output categories instead of probabilities. Soft Voting: Each model provides probability predictions for two categories for the test samples. The final classification result is determined by the decision from the weighted average probabilities. This is suitable when models output probabilities, allowing better utilization of the confidence information.

The Ensemble Voting model adopts the soft voting mechanism mainly for the following reasons: Soft voting can more fully leverage each model's category probability predictions for the test samples. Compared to hard voting, which only considers the quantity of categories, soft voting takes into account the confidence of each model, handling the model's output more flexibly. This allows for a more accurate capture of sample uncertainties, thereby improving overall classification performance. **Algorithm 1** represents the pseudocode of the E-CLEAP model.

```
Algorithm 1. AAC (PseAAC) Feature-based Voting Ensemble Model
Training.
Input: AAC (PseAAC) feature data from the CSV file
Output: Trained voting ensemble model, mean area under the ROC curve
(AUC)
1:
Read the AAC (PseAAC) feature data from the CSV file and shuffle the
data.
2:
Initialize the input file path: inputfile ← 'AAC (PseAAC) features.
csv'
3:
Read the data from the CSV file: data ← pd.read csv()
4:
Shuffle the data: shuffle(data)
5:
Split the data into training set: data train ← data[:int(1 *
len(data)), :]
6:
7:
Prepare the input features and labels for training.
8:
Extract the features: x ← data train[:, 1:19] * 30
9:
Extract the labels: y ← data train[:, 20].astype(int)
10:
11:
Define multiple MLP classifiers and create a voting ensemble model
using these
classifiers.
12:
Define the first MLP classifier: clf1 ← MLPClassifier(. . .)
13:
Define the second MLP classifier: clf2 ← MLPClassifier(. . .)
14:
Define the third MLP classifier: clf3 ← MLPClassifier(. . .)
15:
Define the fourth MLP classifier: clf4 ← MLPClassifier(. . .)
16:
Create a voting ensemble model: model ← VotingClassifier()
17:
18:
```

```
        Perform SMOTE resampling and cross-validation for model evaluation.
19:
Initialize SMOTE:smo ← SMOTE(random_state = 42)
20:
Initialize stratified k-fold cross-validation: cv ← StratifiedKFold()
21:
22:
Initialize mean true positive rate: mean tpr ← 0.0
23:
Initialize mean false positive rate: mean tpr ← np.linspace(0,1,100)
24:
25:
for i, (train,test) in cv.split(x, y) do
26:
        Apply SMOTE resampling: x train, y train      ← smo.fit_resam-
ple()
27:
        Train the model and make predictions: probas_ ← model.fit(x
train,
        y_train).predict proba(x[test])
28:
        Make binary predictions: y pred ← model.predict(x[test])
29:
        Calculate performance metrics:
30:
        Calculate false positive rate, true positive rate, and thresh-
olds: fpr,
        tpr,thresholds ← roc curve(y[test], probas        [:, 1])
31:
        Update        mean        true        positive        rate:
            mean tpr ← mean tpr +
        np.interp(mean fpr, fpr, tpr)
32:
        Set the initial value of mean true positive rate to 0: mean tpr
[0] ← 0.0
33:
        Calculate ROC AUC: roc auc ← auc(fpr, tpr)
34:
        Calculate accuracy: accuracy ← accuracy score(y[test], y pred)
35:
        Calculate recall: recall ← recall score(y[test], y pred)
36:
        Calculate precision: prec ← precision score(y[test], y pred)
37:
        Calculate F1 score: f1 ← f1 score(y[test], y pred)
38:
end for
39:
40:
Calculate the mean ROC curve and area under the curve (AUC).
41:
Divide mean true positive rate by the number of iterations: mean tpr ←
mean tpr / cnt
42:
Set the last value of mean true positive rate to 1: mean tpr[-1] ← 1.0
43:
Calculate mean AUC: mean auc ← auc(mean fpr, mean tpr)
```

**2.3.2 Support Vector Machine (SVM).** SVM has various applications in peptide identification. By optimizing the selection of hyperplanes, SVM can effectively distinguish antigenic and non-antigenic peptide segments. Guo et al. (2008) used SVM models to predict the antigenicity of peptides, providing valuable information for peptide vaccine design [17,18]. Ouellet et al. (2023) applied SVM models to classify peptide segments and successfully differentiated different structural and functional domains [19]. In this study, the radial basis function kernel (RBF) was selected for SVM. The RBF kernel is suitable for nonlinear data distribution and can capture complex boundaries between different classes.

**2.3.3 Naive Bayes classifier (Bayes).** The Bayes model analyzes peptide mass spectra obtained through mass spectrometry techniques, calculates the posterior probability of each possible peptide sequence, and determines the most likely sequence [20]. On the other hand, the Bayes model can predict the epitope positions of unknown peptides, providing a powerful tool for peptide vaccine design and immunogenicity prediction [21]. In this study, the naive Bayes model is mainly used to classify antimicrobial and non-antimicrobial peptide sequences based on AAC features and PseAAC features. The classifier is built by selecting appropriate priors and variance smoothing terms (var_smoothing). In this study, the prior probabilities of the naive Bayes model are estimated based on the class distribution of the training data, and the variance smoothing term is set to 1e-09.

**2.3.4 K-Nearest Neighbors (K-NN).** K-NN classifier is one of the classic algorithms in machine learning and widely used in various research fields due to its relatively simple principles and training process. It calculates the distance between new data and training data, and then selects the k (k≥1) nearest neighbors for classification or regression. The K-NN model determines the classification of unknown peptides by comparing their similarity to known peptides and has been applied in protein identification [22,23]. However, in the case of imbalanced sample sizes, the interpretability and prediction accuracy for rare classes are lower. In this study, a supervised K-NN classifier is chosen, and through multiple adjustments, k = 5 yields the best performance.

**2.3.5 Decision Tree (DT).** DT has multiple applications in peptide identification. In peptide mass spectrometry identification, the decision tree model utilizes the features of peptide mass spectra as branching conditions to progressively divide the feature space of spectra and determine the sequence and modifications of unknown peptides [24]. In peptide sequence classification, the decision tree model assigns unknown peptides to corresponding categories by learning the relationship between known peptide sequences and their features [25]. These application scenarios demonstrate the importance of decision tree models in peptide identification, providing powerful tools for peptide proteomics research and protein analysis. In this study, the decision tree model using the Gini coefficient as the purity measure for nodes is employed.

## 2.4 Experimental setup

A five-fold cross-validation is used to evaluate the training set. The dataset is divided into five equal subsets, with four subsets used for training and the remaining subset used for validation. This process is repeated five times, and the model with the best average performance is selected for the test set. To address the issue of class imbalance, the SMOTE algorithm is applied to oversample the data for each training set partition. Furthermore, parameter optimization is conducted for all experimental models to obtain the best performance. By adjusting hyperparameters such as learning rate, regularization terms, and the number of neurons in hidden layers, the performance and generalization ability of the models are improved.

The independent test set is inputted into the best model, and its performance on unseen data is evaluated based on the model's predictive ability. The model predicts the samples in the

test set and generates corresponding classification results or predictions. This process helps us understand the model's generalization ability in real-world scenarios, i.e., whether it can accurately generalize to unknown data.

We used a standard computer configuration equipped with an i7 10700KF processor and an RTX3060TI-8G AD OC graphics card. Python was used as the primary programming language, along with common scientific computing libraries and machine learning frameworks such as NumPy and Scikit-learn. These tools provided convenience for implementing the experiments and analyzing the results. As for the operating system, we chose the common Windows 10.

## 2.5 Performance evaluation of models

To evaluate the final classification results and facilitate comparison with other models, we used four commonly used metrics in bioinformatics research, including accuracy, recall, precision, and F1-score. The specific formulas for calculating these measurements are as follows:

$$\text{Accuracy} = \frac{TP + TN}{TP + TN + FP + FN} \tag{3}$$

$$\text{Recall} = \frac{TP}{TP + FN} \tag{4}$$

$$\text{Precision} = \frac{TP}{TP + FP} \tag{5}$$

$$\text{Fl}-\text{score} = \frac{2TP}{2TP + FP + FN} \tag{6}$$

Where TN represents true negatives, TP represents true positives, FN represents false negatives, and FP represents false positives.

## 3 Results

Recently, Synthetic Minority Over-sampling Technique (SMOTE) has been widely used as a preprocessing technique to rebalance the proportion of positive and negative samples before constructing the classifier. In SMOTE, in order to prevent information loss, instead of under-sampling the majority class, it performs over-sampling by generating synthetic samples of the minority class from nearest neighbor samples.

To assess whether SMOTE improves antimicrobial peptide (AMP) classification, we incorporated SMOTE as part of the cross-validation procedure. In this process, the training set data is rebalanced by SMOTE, and the constructed classifier is then used to test the samples in the test set.

As shown in **Fig 3**, the E-CLEAP model with SMOTE significantly outperforms the E-CLEAP model without SMOTE, with a notably larger AUC value. Moreover, performance metrics such as Accuracy, Recall, Precision, and F1-score are all superior in the E-CLEAP model with SMOTE compared to the one without SMOTE, as indicated in **Table 1**. Therefore, we consistently utilize the E-CLEAP model with SMOTE in our subsequent data processing.

### 3.1 Results of five-fold cross-validation on the training set

This paper selects 3000 samples as the training set, extracts the first 20 dimensions of AAC and PseAAC features separately, forming a training vector of dimensions 3000×20.

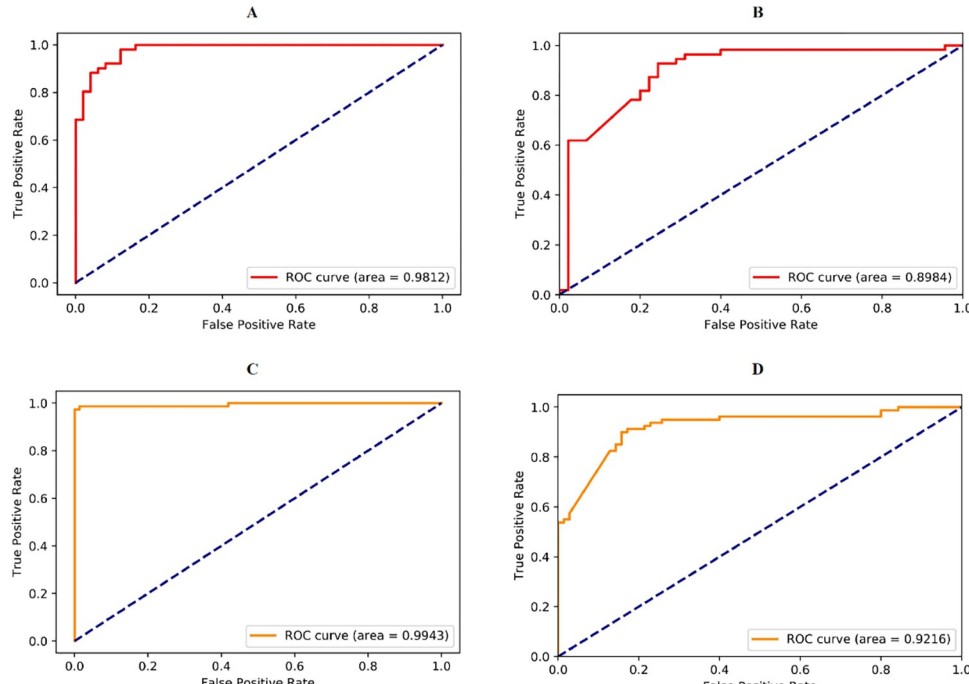

**Fig 3. Performance comparison of the E-CLEAP model before and after applying the SMOTE algorithm (A-E-CLEAP-AAC, B-E-CLEAP-PseAAC, C-E-CLEAP with SMOTE-AAC, D- E-CLEAP with SMOTE-PseAAC).**</ Figure_Caption>

**3.1.1 Results of Amino Acid Composition (AAC) feature.** **Fig 4** displays the ROC curves and AUC values of different models obtained from five-fold cross-validation on the training set using the AAC feature. Our proposed E-CLEAP model achieved the highest AUC value, surpassing the other models. Additionally, the average F1-score of the E-CLEAP model was significantly higher than the other models, with a margin of 0.0139 compared to the second-ranked K-NN model. Although the Precision value of the E-CLEAP model was not the highest, it achieved the highest average accuracy of 0.9173 when the recall rate was 1 (**Table 2**).

**3.1.2 Results of Pseudo Amino Acid Composition (PseAAC) feature.** By performing five-fold cross-validation on the training set using the PseAAC feature, we compared our proposed E-CLEAP model with other models. By observing the average Accuracy, Recall, Precision, and F1-score metrics, we found that the E-CLEAP model achieved an average accuracy of 95.53%. Considering the other metrics, we concluded that the E-CLEAP model performed significantly better compared to the other models, with the K-NN model being the second-best performer and the Bayes model being the least effective (**Table 3**).

**Table 1. Comparison of metrics before and after applying SMOTE algorithm to the E-CLEAP model.**

| Method | Accuracy (%) | Recall (%) | Precision (%) | F1-score (%) |
|---|---|---|---|---|
| E-CLEAP (AAC) | 0.900 | 0.922 | 0.887 | 0.904 |
| E-CLEAP (PseAAC) | 0.760 | 0.618 | 0.919 | 0.739 |
| E-CLEAP with SMOTE (AAC) | **0.973** | **0.987** | **0.962** | **0.974** |
| E-CLEAP with SMOTE (PseAAC) | 0.760 | 0.575 | 0.958 | 0.719 |

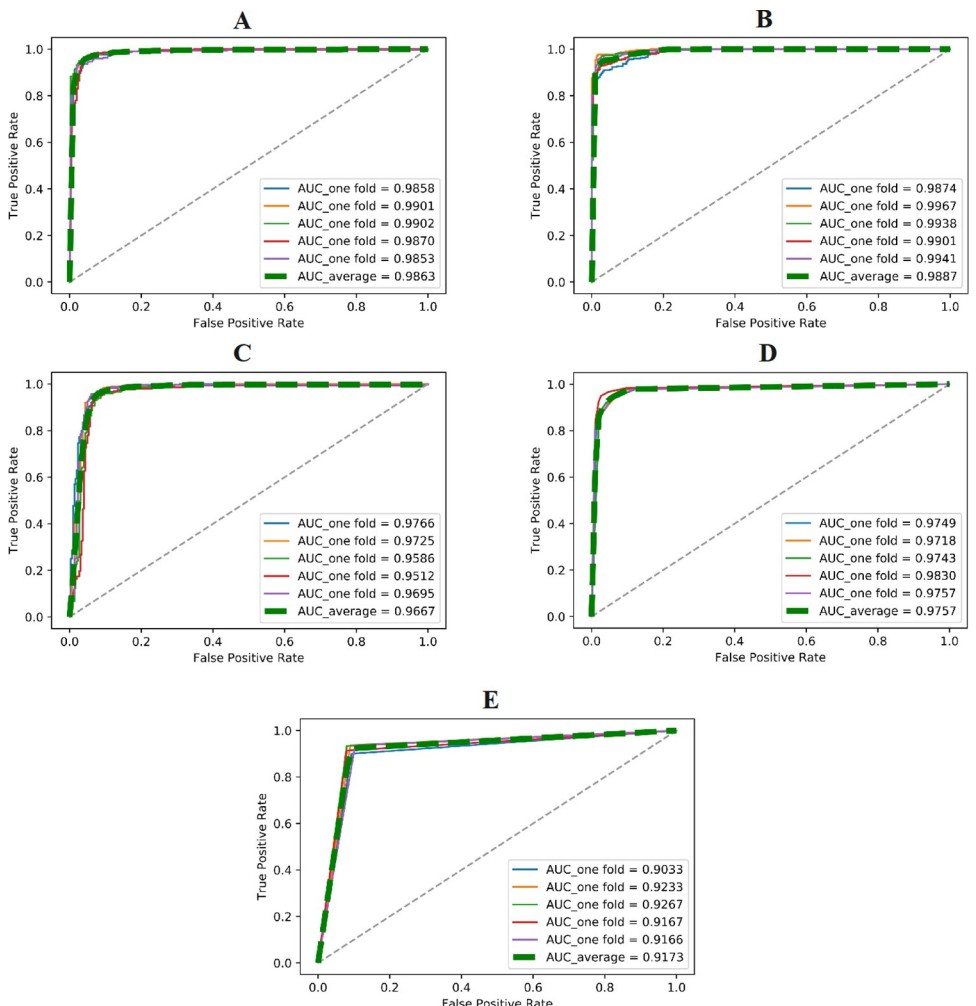

**Fig 4. Validate the performance of different models (A-E-CLEAP,B-SVM,C-Bayes,D-KNN,E-DT) through cross-validation on the training set (AAC features)** .

Fig 5 presents the ROC curves and AUC values of the four classical machine learning methods along with the E-CLEAP model on the training set using the PseAAC feature. Compared to the performance of the E-CLEAP model, the AUC values of the four classical machine learning algorithms based on the training set (PseAAC) are relatively lower. This supports the outstanding generalization and superior ability of the E-CLEAP model in screening antimicrobial peptides.

**Table 2. Comparison of our model with the existing methods through cross-validation on the training set (AAC features).**

| Method | Accuracy (%) | Recall (%) | Precision (%) | F1-score (%) |
|---|---|---|---|---|
| SVM | 93.93±0.79 | 97.60±0.90 | 90.94±1.34 | 94.15±0.74 |
| Bayes | 93.50±0.70 | 93.53±0.87 | 93.48±1.17 | 93.50±0.68 |
| Knn | 94.27±1.39 | 92.73±1.88 | 95.66±1.26 | 94.17±1.43 |
| DT | 91.73±1.11 | 92.40±2.05 | 91.18±1.10 | 91.78±1.16 |
| **E-CLEAP** | **95.53±1.00** | 96.20±1.58 | 94.93±1.02 | **95.56±1.01** |

**Table 3. Comparison of our model with the existing methods through cross-validation on the training set (PseAAC features).**

| Method | Accuracy (%) | Recall (%) | Precision (%) | F1-score (%) |
|---|---|---|---|---|
| SVM | 83.39±2.03 | 97.40±0.90 | 76.11±2.31 | 85.44±1.57 |
| Bayes | 80.76±2.14 | 78.59±2.93 | 82.15±2.03 | 80.32±2.30 |
| Knn | 87.33±1.74 | 93.53±0.48 | 83.23±2.37 | 88.07±1.49 |
| DT | 87.49±2.28 | 93.13±1.84 | 83.71±2.54 | 88.16±2.09 |
| **E-CLEAP** | **87.63±0.86** | 93.13±1.36 | **83.90±1.08** | **88.27±0.81** |

## 3.2 Results on the testing set

We selected 500 samples as the test set, extracting the first 20 dimensions of AAC and PseAAC features separately, forming a test vector of dimensions 500×20.

**3.2.1 Results of Amino Acid Composition (AAC) feature.** To test the specific performance of the E-CLEAP model in antimicrobial peptide recognition, we applied both the E-CLEAP model and four classical machine learning models to identify antimicrobial peptide sequences based on the AAC feature. Firstly, we plotted their ROC curves and calculated the

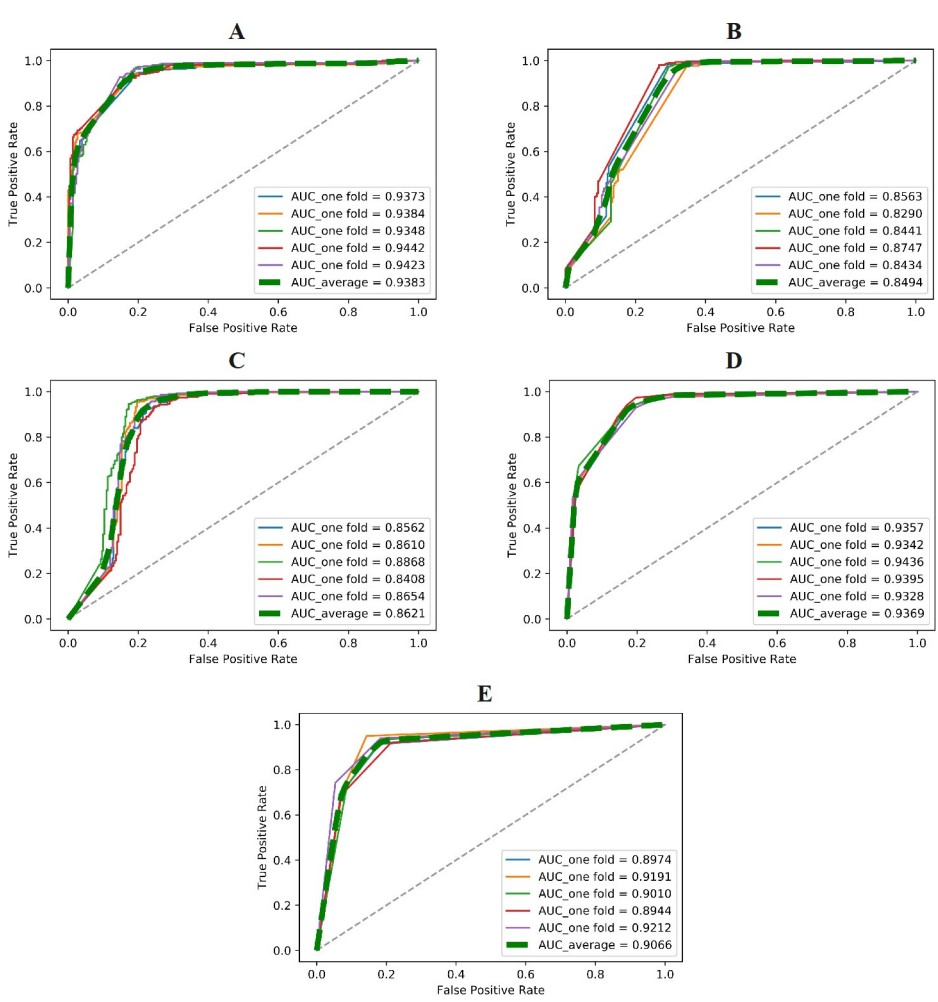

**Fig 5.** Validate the performance of different models (A-E-CLEAP, B-SVM, C-Bayes, D-KNN, E-DT) through cross-validation on the training set (PseAAC features).

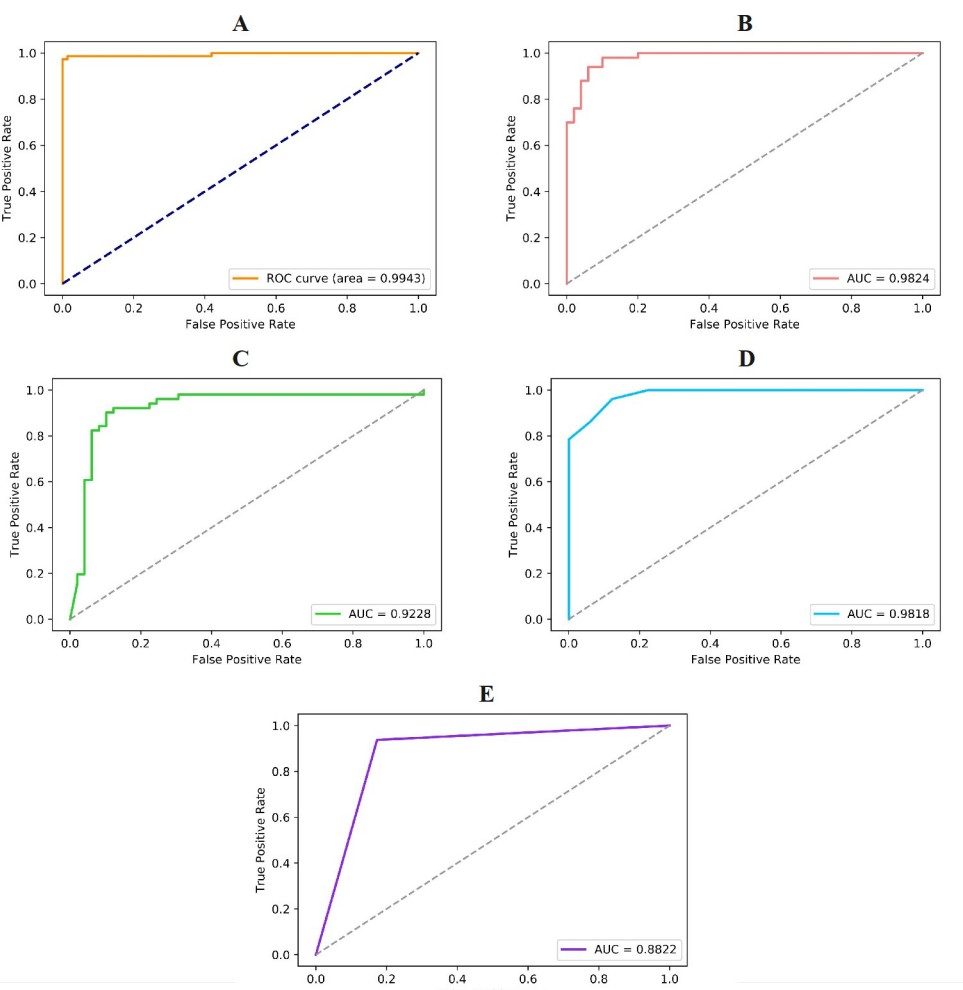

**Fig 6.** Validate the performance of different models (A-E-CLEAP, B-SVM, C-Bayes, D-KNN, E-DT) on the test set (AAC features).

corresponding AUC values, as shown in **Fig 6**. By observing the ROC curves and AUC values in the figure, it is evident that the AUC value of the E-CLEAP model is significantly higher than the other four classical machine learning models, indicating its superior performance in antimicrobial peptide recognition. We can conclude intuitively that the E-CLEAP model performs better in antimicrobial peptide recognition.

To further validate the classification performance of the E-CLEAP model, we also calculated the Accuracy, Recall, Precision, and F1-score for the five models in the antimicrobial peptide recognition task. The relevant results are shown in **Table 4**. From the table, it is

**Table 4. Comparison of our model with the existing methods on the test set (AAC features).**

| Method | Accuracy (%) | Recall (%) | Precision (%) | F1-score (%) |
|---|---|---|---|---|
| SVM | 91.00 | 98.00 | 85.96 | 91.59 |
| Bayes | 88.00 | 86.27 | 89.80 | 88.00 |
| Knn | 90.00 | 86.27 | 93.62 | 89.80 |
| DT | 88.00 | 93.75 | 83.33 | 88.24 |
| **E-CLEAP** | **97.33** | **98.68** | **96.15** | **97.40** |

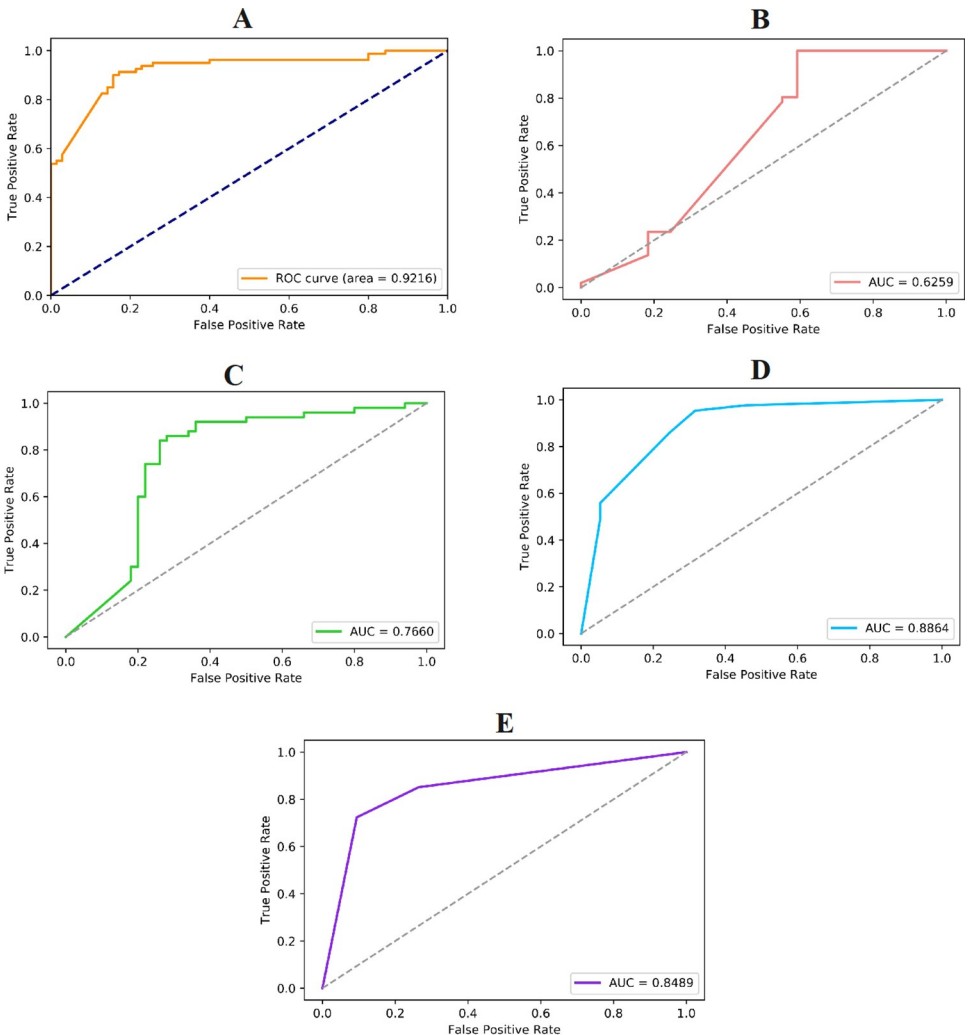

**Fig 7.** Validate the performance of different models (A-E-CLEAP, B-SVM, C-Bayes, D-KNN, E-DT) on the test set (PseAAC features).

evident that the E-CLEAP model outperforms the other models in all the metrics, which further confirms the excellent performance of the E-CLEAP model in antimicrobial peptide recognition.

**3.2.2 Results of Pseudo Amino Acid Composition (PseAAC) feature.** Based on the comparative analysis of the recognition models using the PseAAC feature, the E-CLEAP model exhibits a higher AUC value of 0.9216 (**Fig 7**), and its Accuracy, Recall, Precision, and F1-score metrics are significantly higher than those of the other models (**Table 5**). We can conclude that the E-CLEAP model demonstrates superior performance in the antimicrobial peptide recognition task compared to the other models. These results provide not only intuitive evidence from the perspective of ROC curves and AUC values but also strong support from the comprehensive evaluation of metrics such as Accuracy, Recall, Precision, and F1-score. Our research findings indicate that the E-CLEAP model holds great potential in the field of antimicrobial peptide recognition and provide a solid foundation for further research and application.

**Table 5. Comparison of our model with the existing methods on the test set (PseAAC features).**

| Method | Accuracy (%) | Recall (%) | Precision (%) | F1-score (%) |
|---|---|---|---|---|
| SVM | 63.00 | 80.39 | 60.29 | 68.91 |
| Bayes | 76.00 | 74.00 | 77.08 | 75.51 |
| Knn | 80.00 | 86.05 | 72.55 | 78.72 |
| DT | 79.00 | 85.11 | 74.07 | 79.21 |
| **E-CLEAP** | **84.00** | **87.65** | **83.53** | **85.54** |

### 3.3 Interpreting E-CLEAP using Local Interpretable Model-agnostic Explanations (LIME)

Understanding the biological relevance of the extracted features is challenging. Machine learning models are sometimes referred to as "black box models" due to their complex internal mechanisms. Understanding the contribution of each feature to the model has been considered a challenging aspect of machine learning. SHAP and LIME are commonly used to assess feature importance. LIME and SHAP explore and utilize the characteristics of local interpretability to develop alternative models for black-box machine learning algorithms, providing interpretability.

The E-CLEAP model used in this study consists of four MLP models. Since MLP is a non-tree-based model and the support for the 'shap' library in Python is weaker for non-tree models, the 'lime' library, suitable for interpreting non-tree models, is adopted to evaluate the contribution of extracted features.

Local Interpretable Model-agnostic Explanations (LIME) analysis explains the contribution of individual features to the overall prediction. The assumption of LIME is that every complex model has a linear or explainable relationship in the local space of the dataset. By slightly altering the feature matrix, it is possible to fit a simple model around a sequence. In LIME, a similarity matrix measuring the distance between a query sequence and several permutations is constructed. In **Fig 8**, the interpretation of the prediction results of the E-CLEAP model is shown, presenting a bar chart of the top 6 most important features. Red is highly correlated with antimicrobial peptides, while green is highly correlated with non-antimicrobial peptides.

## 4 Discussion

The aim of this work was to develop an efficient and accurate antimicrobial peptide recognition model. By utilizing different feature extraction methods and machine learning algorithms, we constructed the E-CLEAP model and evaluated it on a large experimental dataset. The highlight of this study is the introduction of the E-CLEAP model and the demonstration of its outstanding performance in antimicrobial peptide recognition.

The remarkable performance of the E-CLEAP model can be attributed to its well-designed model architecture and the utilization of the MLP classifier. Firstly, the E-CLEAP model incorporates multiple features such as the AAC and PseAAC features, capturing the comprehensive information of antimicrobial peptides through a multi-feature fusion strategy. Secondly, the MLP classifier, with its multi-layer structure and non-linear transformation capabilities, can effectively capture the complex features of antimicrobial peptides. The multi-layer structure of the MLP model enables it to learn and represent more intricate patterns, while the non-linear transformations handle the non-linear relationships in antimicrobial peptide classification. Additionally, the MLP classifier offers parameter tunability and optimization capabilities, further enhancing the model's accuracy and stability. In summary, the design of the E-CLEAP

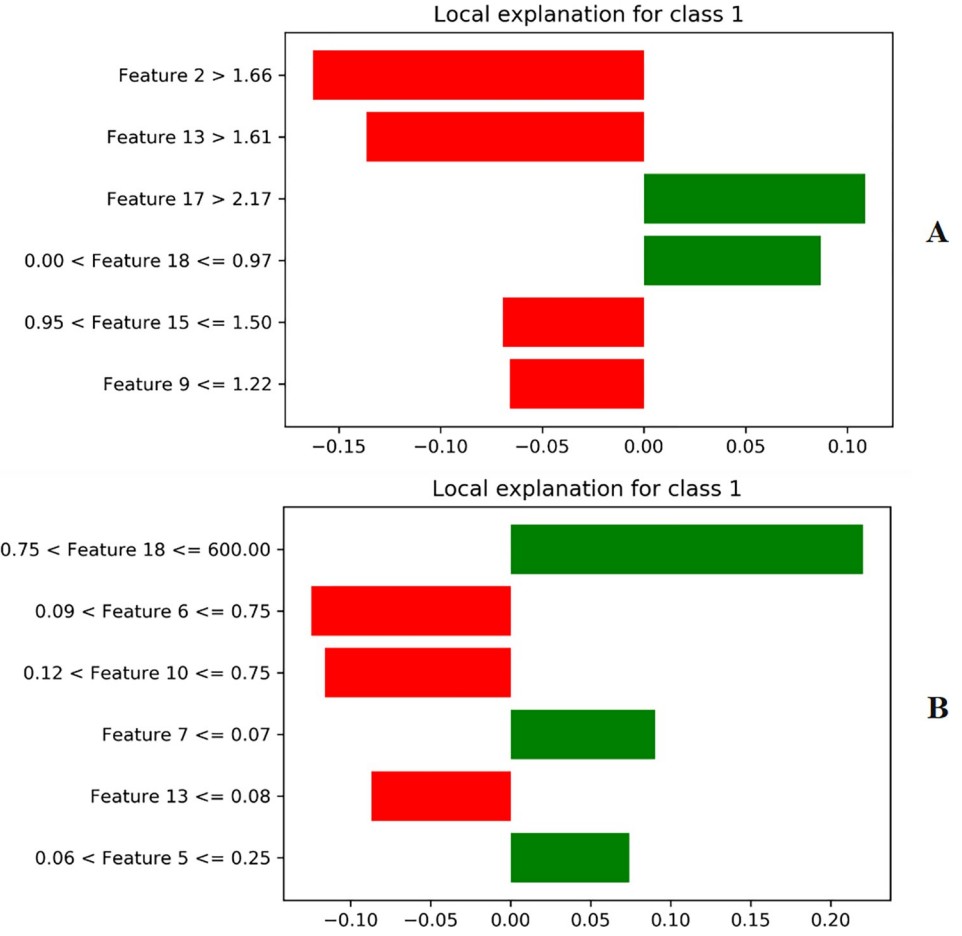

**Fig 8.** LIME interpretable feature contribution plot (A-AAC features,B-PseAAC features).

model and the application of the MLP classifier enable efficient and accurate identification of antimicrobial peptides, with broad prospects for application.

The E-CLEAP model, utilizing four MLP models, exhibits significant advantages over traditional SVM, DT, KNN, and Bayes classifiers in antimicrobial peptide identification tasks. By integrating the predictive results of multiple MLP models, our model accurately captures complex nonlinear relationships, thereby improving predictive performance. Moreover, MLP models demonstrate relative efficiency during training, and E-CLEAP further shortens the training time by averaging the results of multiple MLP models. This provides a viable solution for efficient antimicrobial peptide recognition.

This research holds significant implications for future drug screening. Antimicrobial peptides, as biologically active peptide segments with wide-ranging applications, possess potential antimicrobial properties. By establishing an efficient and accurate antimicrobial peptide recognition model, we can swiftly predict peptide segments with antimicrobial activity in the early stages of drug screening, providing robust support for drug development. The application of the E-CLEAP model can greatly reduce the drug screening cycle and costs, allowing researchers to focus their efforts and resources on peptide segments with higher potential. Furthermore, the E-CLEAP model offers new insights and methods for the discovery of novel antimicrobial drugs. Through in-depth exploration of the features and activities of antimicrobial peptides, new structural and sequence patterns can be discovered, providing guidance for

the design and development of more effective antimicrobial drugs. Therefore, this research holds practical value and significance for future drug screening and antimicrobial drug development, accelerating the drug development process, reducing costs, and providing new opportunities for innovation in the field of antimicrobial drugs.

Despite the achievements made in this work, there are limitations that need to be considered. Firstly, this study lacks experimental validation and is solely based on theoretical modeling. Further experimental verification is necessary to ensure the reliability and stability of the model. Secondly, the dataset used in this study is relatively small, which may limit the generalizability of the model. In future research, efforts should be made to collect larger datasets for validation and improvement. Lastly, this study treats random peptides as negative samples, but they may include some peptides with antimicrobial activity, which could potentially impact the accuracy of the model. In subsequent studies, more accurate negative sample selection and processing methods should be considered to enhance the accuracy and reliability of the model.

## 5 Conclusion

In this study, we proposed an antimicrobial peptide recognition method based on the E-CLEAP model. This method incorporates both the AAC and PseAAC features of antimicrobial peptides and utilizes an MLP classifier to handle the complex features of peptide sequences, thereby improving recognition accuracy and generalization ability. The E-CLEAP model exhibited excellent performance in antimicrobial peptide recognition tasks, outperforming major machine learning models including SVM, Bayes, K-NN, and DT. The E-CLEAP model provides an effective means for the recognition of antimicrobial peptides, enabling fast and efficient identification. This approach offers important support for drug screening, disease research, and the development of biotechnology, and provides new avenues and methods for exploring novel antimicrobial peptide candidates and designing customized antimicrobial peptides. However, this study also has limitations, such as the lack of experimental validation, which limits the reliability of the model in practical applications. In the future, we will further improve the E-CLEAP model by incorporating larger training datasets, introducing more features and information, and conducting experimental validations to enhance the performance and reliability of the model.

## Author Contributions

**Conceptualization:** Si-Cheng Wang.

**Data curation:** Si-Cheng Wang.

**Formal analysis:** Si-Cheng Wang.

**Investigation:** Si-Cheng Wang.

**Methodology:** Si-Cheng Wang.

**Resources:** Si-Cheng Wang.

**Software:** Si-Cheng Wang.

**Validation:** Si-Cheng Wang.

**Visualization:** Si-Cheng Wang.

**Writing – original draft:** Si-Cheng Wang.

**Writing – review & editing:** Si-Cheng Wang.

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
