## [Decision Letter · Decision Letter 0]

19 Jun 2023

PONE-D-23-16655E-CLEAP: An Ensemble Learning Model for Efficient and Accurate Identification of Antimicrobial PeptidesPLOS ONE

Dear Dr. Wang,

Thank you for submitting your manuscript to PLOS ONE. After careful consideration, we feel that it has merit but does not fully meet PLOS ONE’s publication criteria as it currently stands. Therefore, we invite you to submit a revised version of the manuscript that addresses the points raised during the review process.

We look forward to receiving your revised manuscript.

Kind regards,

Shahid Akbar, PhD

Academic Editor

PLOS ONE

Journal Requirements:

"Yes - this study was supported by the China Postdoctoral Science Foundation (2021M691345)"

"NO authors have competing interests"

Reviewers' comments:

Reviewer's Responses to Questions

**Comments to the Author**

1. Is the manuscript technically sound, and do the data support the conclusions?

Reviewer #1: Yes

Reviewer #2: Yes

2. Has the statistical analysis been performed appropriately and rigorously? 

Reviewer #1: Yes

Reviewer #2: Yes

3. Have the authors made all data underlying the findings in their manuscript fully available?

Reviewer #1: Yes

Reviewer #2: Yes

4. Is the manuscript presented in an intelligible fashion and written in standard English?

Reviewer #1: Yes

Reviewer #2: Yes

5. Review Comments to the Author

Reviewer #1: After evaluating the whole manuscript the following comments are needs to incorporated before any possible publication.

1. The literature review of the paper seems really poor, the authors are suggested to add the the latest computational models developed for antimicrobial peptides papers such as, LMPred, AMPlify, ClassAMP, AI4AMP, AmPEP, AMP-GSM,and many more in order to strengthen the literature review section.

2. As the author used ensemble learning model to increase the prediction of the model, however in the introduction section the authors didn't mentioned the effectiveness and motivation to use ensemble learning as provided in iAFPs-EnC-GA and iAtbP-Hyb-EnC models.

3. The authors are suggested to provide a paragraph related to smote oversampling with its proper algorithm.

4. The should provide the results of the model before and after applying smote .

5. The authors should provided the size of the training vector before training a machine learning model.

6. The results of the proposed model should be compared with existing models in terms of predictive results and time complexity.

Reviewer #2: A) In the introduction section the author are advised to clearly mention the novelty of the proposed model.

B) To measure the generalization power and overfitting of the proposed model, an independent dataset is highly mandatory.

C) the ShAp and LIME analysis of the proposed model is required to evaluate the contribution of the extracted features .

D) An extra section are needs to incorporated related to majority voting based ensemble learning.

E) A pseudo code of the proposed model should be added.

F) A github link should be provided in the manuscript regarding the used datasets and the codes of the proposed model.

6. PLOS authors have the option to publish the peer review history of their article (what does this mean?). If published, this will include your full peer review and any attached files.

Reviewer #1: No

Reviewer #2: **Yes: **Dr. Hashim Ali

---

## [Author Response · Author response to Decision Letter 0]

5 Feb 2024

Dear Professor,

Thank you very much for giving us the opportunity to revise the paper. Based on the constructive review comments, we have substantially revised the paper as follows. Thank you again for your guidance.

Yours sincerely,

The Author

Response to Reviewer 1 Comments

1）The literature review of the paper seems really poor, the authors are suggested to add the the latest computational models developed for antimicrobial peptides papers such as, LMPred, AMPlify, ClassAMP, AI4AMP, AmPEP, AMP-GSM, and many more in order to strengthen the literature review section.

Thanks for your comment. I consider this to be a valuable suggestion. Following the reviewer's recommendation, I have included the AI4AMP, AmPEP, AMP-GMS, and AMPlify models to enhance the literature review.

Change made: Page 1, Line 41-49; Page 22-23, Line 349-358

2)As the author used ensemble learning model to increase the prediction of the model, however in the introduction section the authors didn't mentioned the effectiveness and motivation to use ensemble learning as provided in iAFPs-EnC-GA and iAtbP-Hyb-EnC models.

Thank you for your suggestion. I have revised the relevant sentence in line 54 and added references to support this point. I have elaborated on the motivation and effectiveness of using ensemble learning models.

Change made: Page 3, Line 54-58; Page 23, Line 359-364

3)The authors are suggested to provide a paragraph related to smote oversampling with its proper algorithm.

Thank you for your suggestion. I think this is an excellent suggestion. Following the reviewer's advice, we have incorporated additional content regarding SMOTE oversampling and its appropriate algorithms.

Change made: Page 10-11, Line 181-187

4)The should provide the results of the model before and after applying smote.

Thank you for your comment. As you have expressed concerns, the results can be presented more clearly. In accordance with your suggestion, I have added a comparison of the results of the E-CLEAP model before and after using the SMOTE algorithm, along with the inclusion of Figure 3 and Table 1.

Change made: Page 11-12, Line 188-195

5)The authors should provided the size of the training vector before training a machine learning model.

Thank you for your question. The author recognized this deficiency and, following your suggestion, provided the size of the training vector before training the machine learning model.

Change made: Page 12, Line 197-198; Page 15, Line 226-227

6)The results of the proposed model should be compared with existing models in terms of predictive results and time complexity.

Thank you very much for your professional review of the article. In the manuscript, I have provided a detailed comparison between the performance metrics of the E-CLEAP model, including ROC curves, AUC, ACC, and those of traditional SVM, DT, KNN, and Bayes classifiers. In the Discussion section, I have added a new paragraph to clearly articulate the superior performance highlighted by comparing the E-CLEAP model with traditional models.

Change made: Page 21, Line 287-291

Response to Reviewer 2 Comments

1)In the introduction section the author are advised to clearly mention the novelty of the proposed model.

Thank you for your valuable suggestion. I have incorporated the novelty of the proposed model into the introduction section.

Change made: Page 4, Line 64-66

2)To measure the generalization power and overfitting of the proposed model, an independent dataset is highly mandatory.

I highly appreciate your perspective and thank you for your professional review. Regarding the dataset I utilized, I would like to provide the following clarification: the test set is an independent dataset, and during the training process, I further split the training set into training and validation sets for five-fold cross-validation.

3)the ShAp and LIME analysis of the proposed model is required to evaluate the contribution of the extracted features.

Thank you for your suggestion. I consider this to be a valuable suggestion. Following the reviewer's advice, I have incorporated LIME analysis to assess the contribution of features extracted by the E-CLEAP model. Regarding the choice of LIME analysis, I provide the following explanation: Multilayer Perceptrons (MLP) are non-tree models, and the support for the 'shap' library in Python is relatively weaker for non-tree models. Therefore, the 'lime' library is chosen for interpreting non-tree models.

Change made: Page 19-20, Line 255-272

4）An extra section are needs to incorporated related to majority voting based ensemble learning.

Thank you for your suggestion. As suggested by the reviewer, I have added a section in the methodology that provides a detailed explanation of why the soft voting mechanism was chosen and elaborates on the ensemble learning model with a majority voting mechanism. There are two common voting mechanisms, and I believe it is necessary to explain the rationale behind the choice. This adds a level of rigor to the methodology.

Change made: Page 7, Line 114-123

5）A pseudo code of the proposed model should be added.

Thank you very much for your suggestion. As you are concerned, pseudocode can make the model more clearly presented. I have added the pseudocode of my proposed model in the methodology section.

Change made: Page 7-8, Line 123-124

6）A github link should be provided in the manuscript regarding the used datasets and the codes of the proposed model.

I greatly appreciate your professional review of my article. Additionally, I have included a GitHub link in the abstract section, providing access to the dataset used and the code for the proposed model.

Change made: Page 2, Line 23

---

## [Decision Letter · Decision Letter 1]

22 Feb 2024

E-CLEAP: An Ensemble Learning Model for Efficient and Accurate Identification of Antimicrobial Peptides

PONE-D-23-16655R1

Dear Dr. Wang,

We’re pleased to inform you that your manuscript has been judged scientifically suitable for publication and will be formally accepted for publication once it meets all outstanding technical requirements.

Kind regards,

Shahid Akbar, PhD

Academic Editor

PLOS ONE

Additional Editor Comments (optional):

Reviewers' comments:

Reviewer's Responses to Questions

**Comments to the Author**

1. If the authors have adequately addressed your comments raised in a previous round of review and you feel that this manuscript is now acceptable for publication, you may indicate that here to bypass the “Comments to the Author” section, enter your conflict of interest statement in the “Confidential to Editor” section, and submit your "Accept" recommendation.

Reviewer #1: All comments have been addressed

Reviewer #2: All comments have been addressed

2. Is the manuscript technically sound, and do the data support the conclusions?

Reviewer #1: Yes

Reviewer #2: Yes

3. Has the statistical analysis been performed appropriately and rigorously? 

Reviewer #1: Yes

Reviewer #2: Yes

4. Have the authors made all data underlying the findings in their manuscript fully available?

Reviewer #1: Yes

Reviewer #2: Yes

5. Is the manuscript presented in an intelligible fashion and written in standard English?

Reviewer #1: Yes

Reviewer #2: Yes

6. Review Comments to the Author

Reviewer #1: required comments are incorporated i have no further comments and therefore i recommend accept the paper

Reviewer #2: Nil as all the changes are made. no further changes are required. Corrections and grammatical errors are removed as well.

7. PLOS authors have the option to publish the peer review history of their article (what does this mean?). If published, this will include your full peer review and any attached files.

Reviewer #1: No

Reviewer #2: No

---

## [Editor Report · Acceptance letter]

29 Apr 2024

PONE-D-23-16655R1 

PLOS ONE

Dear Dr. Wang, 

I'm pleased to inform you that your manuscript has been deemed suitable for publication in PLOS ONE. Congratulations! Your manuscript is now being handed over to our production team.

Kind regards, 

on behalf of

Dr. Shahid Akbar 

Academic Editor

PLOS ONE